# Use of Measles and Rubella Rapid Diagnostic Tests to Improve Case Detection and Targeting of Vaccinations

**DOI:** 10.3390/vaccines12080823

**Published:** 2024-07-23

**Authors:** Audrey Rachlin, Lee M. Hampton, Paul A. Rota, Mick N. Mulders, Mark Papania, James L. Goodson, L. Kendall Krause, Matt Hanson, Jennifer Osborn, Cassandra Kelly-Cirino, Beth Evans, Antara Sinha, Lenesha Warrener, David Featherstone, David Brown

**Affiliations:** 1Global Immunization Division, Centers for Disease Control and Prevention, Atlanta, GA 30329, USA; 2Division of Viral Diseases, Centers for Disease Control and Prevention, Atlanta, GA 30329, USA; 3Department of Immunization, Vaccines and Biologicals, World Health Organization, 1211 Geneva, Switzerland; 4Bill & Melinda Gates Foundation, Seattle, WA 98109, USA; 5Sound Global Health Consulting, LLC, Seattle, WA 98199, USA; 6Foundation for Innovative New Diagnostics (FIND), 1218 Geneva, Switzerland; 7Gavi, the Vaccine Alliance, Global Health Campus, 1218 Geneva, Switzerland; 8Institute of Global Health, Faculty of Medicine, University of Geneva, 1202 Geneva, Switzerland; 9Public Health Microbiology Division, UK Health Security Agency (UKHSA), London NW9 5EQ, UK; 10Consultant Scientists Ltd., Hastings 4122, New Zealand; 11Laboratory of Respiratory Viruses, Exanthematics, Enteroviruses and Viral Emergencies (LVRE), Oswaldo Cruz Institute, FIOCRUZ, Rio de Janeiro 21040-900, Brazil

**Keywords:** vaccine-preventable disease surveillance, disease elimination, diagnostics, immunization, GMRLN, RDTs

## Abstract

Efforts to control and eliminate measles and rubella are aided by high-quality surveillance data—supported by laboratory confirmation—to guide decision-making on routine immunization strategies and locations for conducting preventive supplementary immunization activities (SIAs) and outbreak response. Important developments in rapid diagnostic tests (RDTs) for measles and rubella present new opportunities for the global measles and rubella surveillance program to greatly improve the ability to rapidly detect and respond to outbreaks. Here, we review the status of RDTs for measles and rubella Immunoglobulin M (IgM) testing, as well as ongoing questions and challenges regarding the operational use and deployment of RDTs as part of global measles and rubella surveillance. Efforts to develop IgM RDTs that can be produced at scale are underway. Once validated RDTs are available, clear information on the benefits, challenges, and costs of their implementation will be critical for shaping deployment guidance and informing country plans for sustainably deploying such tests. The wide availability of RDTs could provide new programmatic options for measles and rubella elimination efforts, potentially enabling improvements and flexibility for testing, surveillance, and vaccination.

## 1. Introduction

Measles is a highly infectious respiratory disease that requires high population immunity to interrupt transmission. Although measles vaccination is estimated to have prevented 57 million deaths worldwide during 2000–2022, an estimated 136,200 individuals died from measles in 2022 and large or disruptive outbreaks were reported in 37 countries [1]. At the same time, reported rubella cases fell to 17,407 globally in 2022 and 98 countries have been verified as having eliminated rubella [2]. The Immunization Agenda 2021–2030 (IA2030) [3] aims to achieve measles and rubella regional elimination targets as a core indicator of vaccine impact and positions measles as a tracer of a health system’s ability to deliver essential childhood vaccines. Efforts to control and eliminate measles and rubella are aided by accurate surveillance data—supported by timely high-quality laboratory testing—to guide decision-making on routine immunization strategies and where to conduct preventive supplementary immunization activities (SIAs) and outbreak response [1,4,5]. Important developments in rapid diagnostic tests (RDTs) for measles and rubella testing, in combination with advances in innovation with molecular detection, present new opportunities for the global measles and rubella surveillance program to greatly improve the ability to rapidly detect and respond to outbreaks. Here, we review the status of RDTs for measles and rubella Immunoglobulin M (IgM) testing, as well as ongoing questions regarding the operational use and deployment of RDTs as part of global measles and rubella surveillance. We also discuss RDT financing challenges in low- and middle-income countries and reflect on the potential ways forward.

## 2. Measles and Rubella Immunization

Vaccines for measles have been in widespread use over the last 50 years, either as a single antigen vaccine or as combination vaccines with rubella, mumps, or varicella antigens [6]. For more than 20 years, the vaccine has been used as part of a global program to control and eliminate measles. The World Health Organization’s Strategic Advisory Group of Experts on Immunization (SAGE) recommends the following: two doses of the measles vaccine administered routinely to all children worldwide—in endemic settings, one at nine months and another at 15-18 months of age; and the rubella vaccine should be administered routinely to all children once the member states they live in can achieve levels of population immunity high enough to minimize the risk of rubella infection in unvaccinated females of child-bearing age [6,7]. Complementing routine immunization against measles and rubella, preventive SIAs help close gaps in population immunity and prevent morbidity and mortality against both diseases. Outbreak response vaccination activities are used to control measles and rubella outbreaks and can also further mitigate measles and rubella morbidity and mortality. For both preventive SIAs and outbreak-response activities, timely and reliable surveillance information—including test results and epidemiological data on whom to target for vaccination—can help to reduce the number of unnecessary vaccinations and can make SIAs more effective by identifying at-risk and unreached groups that need to be vaccinated [5,8].

## 3. Laboratory Testing for Measles and Rubella Surveillance to Guide Immunization Programs

Measles virus infection presents as an illness characterized by rash and fever, with a prodromal phase lasting 2 to 4 days that may include symptoms of cough, coryza, and conjunctivitis [9]. The clinical similarities between measles, rubella, and other diseases that can cause fever and maculopapular rash make laboratory confirmation of suspected cases an essential component of surveillance. Lack of laboratory confirmation may lead to misclassification and inaccurate conclusions, for example, that measles or rubella are not circulating in an area where the viruses are responsible for clinical cases, or misclassification leading to mistaken assessments that measles or rubella are responsible for a significant burden of disease, potentially prompting additional response activities unnecessarily [10].

For a number of vaccine-preventable diseases, the lack of timely and accurate laboratory testing capacity has led to challenges in determining when and where to use available vaccines. For example, if a 2015–2016 yellow fever outbreak in Angola—that subsequently spread to the Democratic Republic of Congo—had been more quickly identified and contained through mass vaccination campaigns, it is estimated that not only would there have been significantly fewer yellow fever cases and deaths, but an additional costly mass vaccination campaign could have been unnessary [11]. For cholera, the large number of recorded outbreaks that have occurred since 2021 has led to unprecedented demand for vaccines from impacted countries. In countries where emergency vaccination campaigns have already been implemented, recurring outbreaks have highlighted the need for improved timeliness and accuracy of testing for identifying areas with new or ongoing transmission, enabling more effective and efficient targeting of outbreak responses. In 2023, the Global Task Force on Cholera Control recommended the expanded use of rapid diagnostic tests to strengthen cholera surveillance, to support more timely and accurate confirmation of cholera cases amidst other diarrheal diseases, and enhance the targeting of preventive and response measures such as vaccination [12,13].

Similar to yellow fever and cholera, timely and accurate diagnostic confirmation for measles and rubella may help to identify the geographic areas that need to be prioritized for vaccination, facilitating more efficient and effective use of vaccines. Diagnostic testing can rule out measles or rubella as the cause of an outbreak, avoiding unnecessary outbreak response vaccination campaigns. Similarly, surveillance data with timely and accurate laboratory confirmation for measles and rubella can enable improved understanding of which age groups or populations need to be targeted during any outbreak or vaccination response [5]. Given how quickly measles can spread, rapid detection and response to measles outbreaks is particularly important to minimize the associated morbidity and mortality [14,15,16].

Accurate and timely surveillance data incorporating laboratory testing results for both measles and rubella can also help monitor the quality of routine immunization services. Increases in cases of measles or rubella can indicate gaps in immunization coverage, including in areas that are otherwise reporting high immunization coverage. For example, an outbreak of measles in Vietnam in 2014 indicated that delays in immunization and lower coverage in specific ethnic groups were creating vulnerabilities masked by high reported overall routine immunization coverage rates [17]. Regular triangulation of vaccine-preventable disease surveillance data with immunization data (e.g., coverage survey data, routine administrative reporting, vaccine stock, etc.) can help identify measles zero-dose or under-immunized children and prompt efforts to vaccinate them [18].

## 4. Measles and Rubella Laboratory Testing

### 4.1. Current Laboratory Testing Methods

Laboratory-based enzyme immunoassay (EIA) testing of sera for measles and rubella immunoglobulin M (IgM) antibodies is currently the mainstay of testing for both diseases [19]. Measles and rubella EIA testing at national reference laboratories is highly sensitive and specific when performed correctly but has a lower positive predictive value in low-incidence settings [20]. Alternative samples, such as capillary blood and oral fluid that can be tested immediately or collected as dried blood spots, can also be used for IgM antibody detection but they have not been widely adopted for measles and rubella surveillance [10].

Over recent years, quantitative real-time reverse-transcriptase polymerase chain reaction (qRT-PCR) has been used increasingly to complement IgM detection for case confirmation [19]. qRT-PCR has a high sensitivity when specimens are collected soon after the onset of a rash. Other clinical specimens can also be used for quantitative real-time or conventional RT-PCR, including throat or nasal swabs, oral fluid, urine, and capillary blood. Detection of ribonucleic acid (RNA) by RT-PCR also has the added advantage of enabling genotyping, which can be used for molecular epidemiology [21,22].

### 4.2. Role of the Global Measles and Rubella Laboratory Network

The World Health Organization (WHO) Global Measles and Rubella Laboratory Network (GMRLN), a network of over 700 laboratories operating in 191 countries [23,24], has played a key role in supporting laboratory testing worldwide, including evaluating commercially available measles and rubella EIA test kits. These GMRLN evaluations have resulted in several EIA kits being validated for use in network laboratories. GMRLN laboratories typically receive more than 270,000 samples per year for measles and rubella diagnostic serology from all countries [1,25]. Genotyping and sequencing of measles and rubella viruses also play an increasingly important role in all WHO regions in tracking virus transmission pathways [26]. Virologic surveillance of circulating measles viruses is used to monitor progress toward elimination and GMRLN members update the online Measles Virus Nucleotide Surveillance (MeaNS) and Rubella Virus Nucleotide Surveillance (RubeNS) database once isolates are sequenced [25,26,27].

### 4.3. Laboratory Testing Challenges

Despite these achievements, challenges related to timely and accurate laboratory testing persist, primarily due to limited laboratory and staff capacity and the availability of EIA test kits [28]. In areas with minimal infrastructure or remote locations, these challenges are often intensified by insufficient specimen collection, storage, transport capacity, and laboratory accessibility [29,30]. Blood and other specimens from suspected measles and rubella cases must be carefully transported from the point of collection to the relevant reference laboratory. In hard-to-reach areas, this adds substantially to the time needed to complete testing and may lead to specimens not arriving or being in poor condition if they are not collected, stored, and transported properly [25]. For example, in Nigeria during 2023, only 60.6% of reported measles cases had a blood specimen collected for testing, and among those specimens sent to the lab, only 35.2% arrived within 3 days of collection, falling well below the target for both of these indicators of surveillance system performance of 80% or above [31]. Given these challenges, decentralized testing for measles and rubella, using simplified diagnostic tools, could potentially have significant benefits on the timeliness and utility of measles and rubella diagnostic testing and support timely outbreak-response SIAs.

## 5. Current Status of Measles and Rubella IgM Rapid Diagnostic Tests

### 5.1. Measles IgM Rapid Diagnostic Tests

Rapid diagnostic tests (RDTs) based on lateral flow technology are increasingly used for the diagnosis of infections by detection of antibodies or antigens (e.g., malaria, COVID-19, HIV). These tests can be performed at ambient temperature and under field conditions without the use of complex electrical equipment, eliminating the need for reverse cold chain transport of specimens, and with minimal technical training requirements [32,33]. Researchers at the UK Health Security Agency (UKHSA, formerly Public Health England [PHE]) developed an RDT for measles IgM antibodies in 2011 [34]. The test was evaluated using 170 serum specimens collected through measles surveillance or vaccination programs in Ethiopia, Malaysia, and the Russian Federation. An additional 282 oral fluid specimens from the measles, mumps, and rubella (MMR) surveillance program of the United Kingdom of Great Britain and Northern Ireland were also screened, with the Microimmune measles IgM capture EIA (Microimmune Ltd., Hounslow, England) as the gold standard for comparison [34]. With serum specimens, the RDT had a sensitivity of 91% (69/76) and a specificity of 94% (88/94); with oral fluid, the sensitivity was 90% (63/70) and the specificity was 96% (200/208) [34].

The study also demonstrated that the RDT could support measles virologic surveillance, as viral RNA could be extracted from test devices, amplified by standard RT-PCR, and used for sequence analysis to determine the genotype at national or regional laboratories. Here, a panel of 24 oral fluids was used to investigate whether measles virus haemagglutinin (H) and nucleocapsid (N) genes could be amplified by PCR directly from used RDTs. Both H and N genes were reliably detected and the N genes could be sequenced for genotyping. Measles virus genes could be recovered from RDT test strips after storage for five weeks at 20–25 °C [34]. A subsequent evaluation of 103 suspected measles cases during a 2010 nationwide measles outbreak in Zimbabwe confirmed previous findings that it was possible to amplify measles genes from oral fluid using RDT test strips and identify the infecting measles genotype and strain. During the evaluation, there was no loss in the sensitivity of detection of the measles H gene from RDT strips compared to direct detection from oral fluid samples [35].

With support from the Bill & Melinda Gates Foundation (BMGF), this prototype RDT was redesigned into a more robust, cassette-housed device, requiring fewer procedural steps to generate a result, with demonstrated high sensitivity (95%) and specificity (96%) when evaluated on stored sera against the Enzygnost^®^ anti-measles virus IgM EIA reference assay (Siemens Healthcare GmbH, Marburg, Germany) [36,37]. In unpublished field trials from India and Uganda, the sensitivity of measles IgM detection in capillary blood using the RDT was comparable to published performance with RDTs and to current laboratory-based EIA testing; the sensitivity was slightly lower using oral fluid [37].

In a subsequent 2019 field trial in Malaysia, the sensitivity of the measles IgM RDT was lower than reported in earlier evaluations and measles endemic settings. RDT sensitivity using capillary blood was 43% and specificity was 98%; using oral fluid, sensitivity was 26% and specificity was 97% [38]. However, parallel measles and rubella vaccination and mop-up campaigns that occurred within the 8–56 days preceding specimen collection may have led to potential vaccine responses in some cases, contributing to the discordant results. Additionally, measles incidence in Malaysia decreased between the study’s design and implementation phases, ultimately resulting in decreased statistical power. These data highlight the need to further evaluate both the performance and potential role of IgM RDTs in low-incidence elimination settings [38].

BMGF and Médecins Sans Frontières have provided further support for a partnership with UK-based Global Access Diagnostics as well as Senegal-based Diatropix, a company associated with the Pasteur Institute in Dakar, to develop a commercially available version of the UKHSA RDT [39,40]. Because of the lower costs and greater ease of use of such lateral flow RDTs—which are similar in operation to home pregnancy tests, COVID-19 at-home tests, and HIV RDTs—their implementation could be scaled up for use in decentralized settings in many countries and used by a wide range of healthcare workers with limited laboratory training.

### 5.2. Rubella IgM Rapid Diagnostic Tests

In contrast to measles IgM RDTs, multiple rubella IgM RDTs are already commercially available, although none have yet been validated by the GMRLN or the WHO. These tests support the diagnosis of recent or active rubella infection, as well as the determination of immune status. Because of the TORCH (toxoplasmosis, others [syphilis, hepatitis B], rubella, cytomegalovirus, and herpes simplex) panel screening among pregnant women and infants in high-income and upper-middle-income countries [41,42], the existing market demand for such tests is substantial. The logistical and operational concerns that make measles IgM RDTs attractive also apply to rubella IgM RDTs. In countries that have not eliminated rubella, pairing measles and rubella IgM RDTs (preferably within a single combined testing platform) would support parallel or sequential measles and rubella testing, which is the standard for EIA testing.

### 5.3. Integration of RDTs into Measles and Rubella Surveillance Programs

Accurate measles and rubella IgM RDTs could eventually create a range of options for more decentralized testing for measles and rubella. For example, an RDT that could test oral fluid and capillary blood specimens would expand who could collect samples and reduce the supplies needed to collect them [37]. Oral fluid and capillary blood specimens are less invasive than whole blood samples and can be collected with less training and equipment than venipuncture. A less-invasive test could potentially result in greater patient testing compliance, therefore improving the representativeness of surveillance [43]. When combined with the simplicity of performing RDTs, a wide range of front-line healthcare workers could utilize such tests. Such characteristics make measles and rubella IgM RDTs particularly attractive for testing in areas in which establishing reference laboratories and transporting specimens are difficult, such as many low- and middle-income countries, Pacific Island countries, areas with civil unrest, and remote, hard-to-reach populations.

The results obtained from measles and rubella IgM RDTs could potentially be integrated into countries’ testing systems at different administrative levels, enabling more rapid detection of and timely response to measles cases and outbreaks. For example, RDTs could be used to rapidly expand subnational laboratory networks without the need to equip the laboratory for EIA or qRT-PCR. The GMRLN has a strong training program that could be expanded to include training on the use of RDTs in subnational laboratories. Specimens collected at remote locations and tested by RDTs would be submitted to national laboratories for confirmatory testing by EIA. RDTs with positive results would then be referred to national laboratories or regional laboratories for genotyping or sequencing, as needed. To ensure high-quality standardized testing in subnational laboratories, implementing RDTs in parallel with quality management systems and assurance programs will help to reinforce the validity of the test results, build capacity for the use of RDTs for surveillance, and ensure alignment with global guidance for the post-market surveillance of new diagnostic tools. National labs will likely have a large role to play in the quality management and assurance of RDT use in subnational laboratories if they are to be introduced within a country.

Another option for expanded laboratory diagnostics would be for district surveillance officers to use such RDTs to test suspected measles cases during field investigations. Thorough integration of results from RDT testing into routine health information management systems at both the local and national levels would also help to ensure that testing results are reported and available to guide public health decisions at all administrative levels. This would likely require expansion of the number of variables collected in routine datasets, including RDT usage and results, as well as the strengthening of electronic data reporting in many countries where paper systems are still the current norm.

## 6. Deployment of Measles and Rubella IgM Rapid Diagnostic Tests: Challenges in Low- and Middle-Income Countries

### 6.1. Financing Challenges for Measles and Rubella IgM RDT Deployment

To realize the potential benefits of measles and rubella IgM RDTs to make measles and rubella surveillance in low- and middle-income countries more effective and efficient, such RDTs would need to be deployed at scale in those countries. That deployment would need to balance the timeliness and detail of the data collected with the costs of ongoing distribution and use of the tests so that results can be available to guide decision-making as sustainably as possible. One of the main challenges affecting the development of RDTs for high-priority diseases is the lack of adequate financing. In general, global funding for diagnostics is limited and the lack of incentive from the private sector has been a challenge for the development of diagnostic tests [44,45,46].

In the long term, RDT use would ideally be funded and supported by government healthcare, public health systems, and private providers. However, in the short term, international groups such as Gavi, the Vaccine Alliance (GAVI) are likely to be major funders for measles and rubella RDT procurement in low- and middle-income countries. Although GAVI is primarily focused on improving the availability of vaccines to low- and middle-income countries and supporting countries in delivering vaccines to their target populations, it has also set up a diagnostic test support program to improve the availability of fit-for-purpose diagnostic tests to make targeted vaccination programs against diseases like measles and rubella more effective, efficient, and equitable [47]. GAVI published a measles IgM RDT market-shaping roadmap describing its expectations on how such a test might be introduced to the market and the potential volume of demand for such a test and is working on a similar rubella RDT roadmap [46]. While GAVI can provide “pull” market incentives (i.e., strategies in which organizations aim to increase the demand for a product and draw (“pull”) consumers to the product) to encourage manufacturers to produce diagnostic tests and can fund work to validate test kits and to develop guidelines on how to best use new tests, its procurement funding support is currently limited to the 54 countries eligible for GAVI’s support (Figure 1) [48]. As a result, international funding support for the deployment of measles and rubella IgM RDTs outside of those countries will need to come from other sources. Since many countries experiencing large or disruptive measles outbreaks are not eligible for GAVI procurement funding support for diagnostic tests [49], they would likely benefit from funding support from other sources for introducing RDTs.

A further limitation on widespread deployment is the lack of funding to support rapid development of quality RDTs. GAVI’s funding for the purchase of RDTs is applicable as a future incentive but does not directly support the research and development efforts needed to bring RDTs into the market. International funders will be essential to close the funding gaps and ensure timely development of high-performing, tailored, cost-effective tests and their availability for procurement and implementation by GAVI and other partners.

### 6.2. Validation of Commercially Available Measles and Rubella IgM RDTs and Guidance on Deployment

Validation of diagnostic tests, i.e., evaluation by an independent organization of the accuracy and reliability of a diagnostic test, is essential to indicate to international organizations, national governments, and others involved in diagnostic test procurement which tests are worth procuring and deploying. The GMRLN has for years independently assessed measles and rubella IgM EIA tests and identified those with acceptable performance characteristics for use by national, regional, and global reference laboratories throughout the network. Such assessments have drawn on data from the independent laboratory testing of test kits as well as reviews of test performance and manufacturing quality assurance provided by test producers [23,24,25]. The GMRLN is positioned to play a similar role for measles and rubella IgM RDTs, helping to guide international organizations, including GAVI, on whether specific RDTs warrant scaling up. The Foundation for Innovative New Diagnostics (FIND), together with the GMRLN and the WHO, have recently published target product profiles (TPPs) for measles and rubella IgM RDTs and EIAs [50]. These TPPs describe the improvements necessary for the development, evaluation, and scale-up of diagnostic tests and will help to indicate to manufacturers the characteristics RDTs will need to have to potentially be procured with funding from international organizations [50].

The development of guidance on RDT deployment can also benefit from a combination of modeling studies and pilot projects. Pilot projects in Uganda, India, and Malaysia with prototype measles IgM RDTs have already demonstrated that such RDTs can be readily used by healthcare workers. For example, during a 2019 field trial in Malaysia, an evaluation team including co-investigators from the Malaysian MOH, U.S. CDC, UKHSA, and WHO implemented a pilot of measles IgM RDT testing in 34 health clinics following standardized one-day regional training. After this one-day training, agreement between readers of the RDTs in health clinics was very high (99% for capillary blood and 97% for oral fluid). A knowledge, attitude, and practices (KAP) survey documented the extensive (92%) previous experience of clinic staff with other RDTs, which was likely an important contributor to the success of using the measles RDT in Malaysian clinics. Furthermore, knowledge was very high (91%) after the formal training and remained high (80%) ≥9 months later among survey respondents. Although not statistically significant, the number of days until public response activities were initiated (e.g., active case detection, mop-up vaccination) decreased by almost five days during the same study, likely related to receiving the RDT results at the point-of-care compared to the time required to perform an EIA in the national laboratory and report results [37,38]. Future pilot studies supported by GAVI in Ethiopia, Zambia, and Nepal will focus on the operational aspects of deploying measles RDTs, and modeling studies are also underway that will assess the tradeoffs between deployment costs and the speed of detection of cases and outbreaks with different strategies for measles and rubella RDTs.

Data on the cost-benefit and effectiveness of measles and rubella RDTs are also needed to make further investment decisions regarding diagnostics development to provide an economic case for future investment in RDTs. There have been estimates of the costs of RDTs for various diseases such as HIV, malaria, and tuberculosis [51,52,53] that have indicated the decentralization of testing using RDTs is likely to be cost-saving or cost-effective in resource-limited settings [54]. The fast turnaround time of RDTs may also result in a reduction in unnecessary clinical treatment costs, leading to savings for both patients and clinics [55].

While the previous literature has also documented the costs of various aspects of measles and rubella surveillance [56], there is limited information available regarding the costs of implementing measles and rubella RDTs as part of a national surveillance program. Such data would help inform decision-makers and public health planners on the magnitude of resources required for introduction, how these costs may vary across geographies (e.g., rural vs. urban, endemic vs. elimination settings), and how this could impact the public health response during measles and rubella outbreaks. Figure 2 presents the potential benefits and challenges to the introduction and use of RDTs in various settings. The relative benefit of use in these different settings is likely to vary by country and within a country. One benefit is that capillary blood and oral fluid samples could be used at community and health clinic levels, improving patient compliance with testing [37]. However, implementation at these lower levels of the health system would also require reprioritization and optimization of current resources in terms of training, supply logistics, quality control and assurance, and processes for reporting results [32,37,57,58].

### 6.3. Future Directions and Way Forward

Once the GMRLN has preliminarily identified a measles IgM RDT(s) that meets deployment criteria, the GAVI secretariat is planning to fund pilot projects in multiple countries eligible for GAVI’s new vaccine support to assess the costs and effectiveness of different deployment strategies, e.g., placing RDTs in healthcare facilities that would treat suspected measles cases, with district surveillance officers that would investigate such cases. Stocking tests in many sites might speed up the detection of cases and outbreaks while also providing a market incentive to manufacturers to ensure they continue to produce high-quality tests. Without this, given the small commercial market for RDTs, there is a risk that the products will be discontinued or will not be put forward for Stringent Regulatory Authority (SRA) approval. However, this approach may also markedly increase the costs of procuring enough tests to maintain stocks at each site and training staff on their use, so clear information on the benefits and costs of these options will be critical for shaping deployment guidance and informing country plans for sustainably deploying such tests. Once the tests are in use in countries more broadly, careful evaluations of the countries’ experiences with deployment can inform further revisions in deployment guidance, country support needed, and how countries implement such tests.

## 7. Conclusions

Measles and rubella IgM RDTs have the potential to markedly improve testing and surveillance for measles and rubella, especially for rapid detection and response to measles and rubella outbreaks. They can potentially contribute to improved effectiveness and efficiency of measles and rubella elimination efforts and vaccination strategies, particularly in low- and middle-income countries. Efforts to develop and validate measles and rubella IgM RDTs that can be produced at scale are underway. International funding will be critical to ensuring the timely development of high-performing, cost-effective tests. Once such validated RDTs are available, funding support will also be needed from international organizations and countries to deploy them. However, since the benefits of use are likely to vary by country and within a country, their implementation will need to carefully balance the costs of ongoing distribution and use so that test results can be available to guide decision-making as sustainably as possible.

## Figures and Tables

**Figure 1 vaccines-12-00823-f001:**
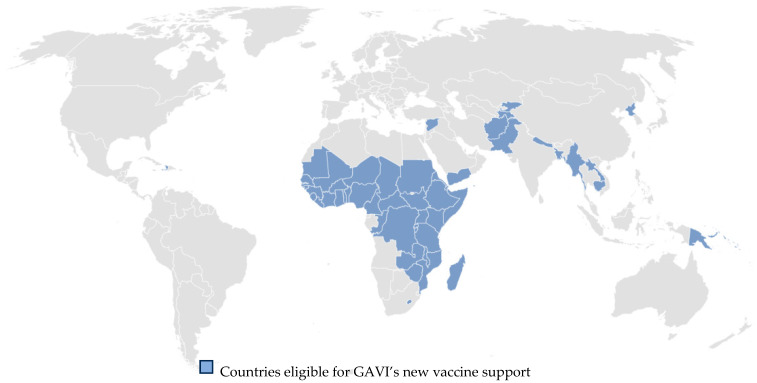
Countries eligible for GAVI’s new vaccine support in 2024 [48].

**Figure 2 vaccines-12-00823-f002:**
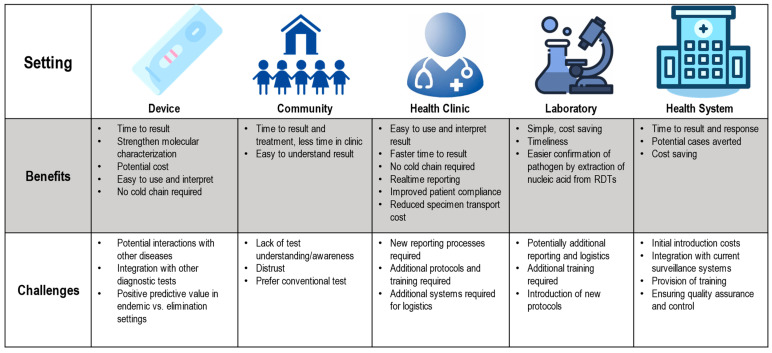
Potential benefits and challenges associated with the introduction of measles and rubella RDTs in various healthcare settings [37,57].

## Data Availability

No new data were created or analyzed as part of this report. Data sharing is not applicable to this article.

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
