# Peer review of "Use of Measles and Rubella Rapid Diagnostic Tests to Improve Case Detection and Targeting of Vaccinations"

_vaccines, 2024, doi:10.3390/vaccines12080823_

Round 1

Reviewer 1 Report

Comments and Suggestions for Authors

A comprehensive and timely review of RDTs for measles and rubella. Clearly explains the development and potential role of these tests in the efforts to eliminate measles and rubella. An important contribution to this subject and is likely to be often referenced.

Lines 71-73. What is meant by 'timely and reliable information'? In the context of this paragraph, test results should be stated, rather than assumed. However, if it is 'test results' stated how do these test results identify at-risk and unreached groups that need to be vaccinated? Perhaps replace 'timely and reliable information' with 'timely and reliable test results and epidemiological data'? Or 'timely and reliable surveillance information, including test results and epidemiological data,'?

Lines 156-163. Repetition. Suggest deleting.

Lines 200-203. National labs will have a large role to play in the use of RDTs if they are to be introduced to testing systems within a country. This is hinted at here and in lines 287-290, also figure 2. However, the role of national labs in providing appropriate quality management systems/assurance should be more clearly stated and defined in the text in one or two sentences, probably in the paragraph of lines 194-203.

Line 221. Delete 'as Gavi'.

Author Response

Reviewer 1

Dear Reviewer,

Thank you for your thoughtful review of our manuscript entitled “Use of Measles and Rubella Rapid Diagnostic Tests to Improve Measles and Rubella Detection and Targeting of Vaccination”, submitted to the Vaccines Special Issue “A World without Measles and Rubella: Meeting the Regional Elimination Targets on the Path to Global Eradication”. We appreciate the time and effort you have dedicated to providing this helpful feedback. Below, we address each of your comments and suggestions:

Comments and Suggestions for Authors

A comprehensive and timely review of RDTs for measles and rubella. Clearly explains the development and potential role of these tests in the efforts to eliminate measles and rubella. An important contribution to this subject and is likely to be often referenced.

Lines 71-73. What is meant by 'timely and reliable information'? In the context of this paragraph, test results should be stated, rather than assumed. However, if it is 'test results' stated how do these test results identify at-risk and unreached groups that need to be vaccinated? Perhaps replace 'timely and reliable information' with 'timely and reliable test results and epidemiological data'? Or 'timely and reliable surveillance information, including test results and epidemiological data,'?

Thank you for this suggestion. We agree there is a need to clarify the meaning of timely and reliable information. To lines 72-76 we have added “For both preventive SIAs and outbreak response activities, timely and reliable surveillance information, including test results and epidemiological data on whom to target for vaccination, can help to reduce the number of unnecessary vaccinations and can make SIAs more effective by identifying at-risk and unreached groups that need to be vaccinated”.

Lines 156-163. Repetition. Suggest deleting.

Thank you for this comment. We have removed the duplicate text (now lines 202-208).

Lines 200-203. National labs will have a large role to play in the use of RDTs if they are to be introduced to testing systems within a country. This is hinted at here and in lines 287-290, also figure 2. However, the role of national labs in providing appropriate quality management systems/assurance should be more clearly stated and defined in the text in one or two sentences, probably in the paragraph of lines 194-203.

We agree that it is important to describe the role of national labs in providing appropriate quality assurance and control. To lines 253-259 we have added “To ensure high quality standardized testing in subnational laboratories, implementing RDTs in parallel with quality management systems and assurance programs will help to reinforce the validity of the test results, build capacity for the use of RDTs for surveillance, and help to ensure alignment with global guidance for the post-market surveillance of new diagnostic tools. National labs will likely have a large role to play in the quality management and assurance of RDT use in subnational laboratories if they are to be introduced within a country”.

Line 221. Delete 'as Gavi'.

Thank you for this comment. As Gavi, the Vaccine Alliance is the official name for GAVI, which we use throughout the remainder of the manuscript, we have kept this in. However, we realized “Gavi” should have been capitalized as “GAVI” and have replaced this throughout the subsequent text.

Thank you once again for your valuable feedback. We hope the revisions address your concerns. We have also attached the revised tracked version of the manuscript for you to view.

Best regards,

Audrey Rachlin, PhD, MSc

Epidemiologist

Global Immunization Division

Centers for Disease Control and Prevention (CDC)

Reviewer 2 Report

Comments and Suggestions for Authors

Measles and rubella are exanthematous virus infections that cause considerable morbidity and mortality worldwide, despite the availability of effective immunisation for close to 50 years. Contributory factors are the reduced uptake of immunisation due to individual choice or due to disruption of national immunisation programmes. Measles and rubella present with coryzal symptoms, fever, conjunctivitis and a maculopapular skin rash, that can be difficult to differentiate on clinical grounds alone from other viral diseases presenting with similar clinical features. Therefore, laboratory testing to confirm the diagnosis is important to monitor the effectiveness of routine immunisation programmes and also implement supplementary immunisation activities to deal with outbreaks.

Currently, laboratory-based enzyme immunoassay testing of sera for measles and rubella IgM antibodies is the mainstay of testing, but there are logistical problems in achieving these results in a timely manner. Genotyping of these two viruses are also available but have similar logistical disadvantages. Lateral  flow technology-base, rapid diagnostic tests have made a huge impact on the control of malaria, COVID-19 and HIV. Similar approaches would be highly valuable for the control of measles and rubella. There are commercial rapid diagnostic tests for detecting rubella IgM but subject to limited validation. 

This report highlights the development of rapid diagnostic tests for anti-measles IgM which appears to have promise. This communication cites some information about the currently available validation of rapid diagnostic tests to detect measles IgM. However, this manuscript does not provide a systematic review of the available rapid diagnostic tests for measles and rubella and their reliability and effectiveness under field conditions. 

“Here, we review the status of RDTs for measles and rubella Immunoglobulin M (IgM) testing, as well as ongoing questions regarding operational use and deployment of RDTs as part of global measles and rubella surveillance. Researchers at UK Health Security Agency (UKHSA, formerly Public Health England

[PHE]) developed an RDT for measles IgM antibodies in 2011 [29]. With serum specimens, the RDT had a sensitivity of 91% (69/76) and a specificity of 94% (88/94); with oral fluid, the sensitivity was 90% (63/70) and the specificity was 96% (200/208) [28]. With support from the Bill & Melinda Gates Foundation (BMGF), this prototype RDT was redesigned into a more robust, cassette-housed device, requiring fewer procedural steps to generate a result, with demonstrated high sensitivity (95%) and specificity (96%) when evaluated on stored sera against the Enzygnost® anti-measles virus IgM EIA reference assay (Siemens Healthcare GmbH, Marburg, Germany) [30]. Additional studies have demonstrated that RDTs can also support measles virologic surveillance, as viral RNA can be extracted from test devices, amplified by standard RT-PCR and used for sequence analysis to determine genotype at national or regional laboratories [31]. 

However, in a 2019 field trial in Malaysia, the sensitivity of the measles IgM RDT was lower than reported in earlier evaluations and measles endemic settings. RDT sensitivity using capillary blood was 43% and specificity was 98%; using oral fluid, sensitivity was 26% and specificity was 97%, highlighting the need to further establish the role of the device in elimination settings.”

From the information given, the reader cannot judge the quality of the evidence provided, without having to review the original manuscripts cited. For example, there is no critical analysis of why the 2019 field trial in Malaysia had poor sensitivity using the rapid diagnostic test for anti-Measles IgM and whether since that time efforts have been made to improve the performance of these assays in under field conditions. 

It is not clear what additional contribution to this topic is made in this manuscript compared to the cited paper from 2020: https://doi.org/10.1016/j.coviro.2020.05.007.

This manuscript appears to have two objectives. First, is to critically review the current scientific validity of rapid diagnostic tests for measles and rubella. Secondly, it tries to make a health-economic case for employing these techniques for measles and rubella surveillance on a global scale. In my opinion, it does not achieve either of these objectives, sufficiently to justify publication in the current format . The options for improving this manuscript are to either carry out a systematic review of currently available conventional and rapid diagnostic tests for measles and rubella, possibly in comparison with other similar viral illness. An alternative is to carry out a review of the health-economic case for using rapid diagnostic tests which could be achieved in a much shorter manuscript length.

Author Response

Reviewer 2

Dear Reviewer,

Thank you for your thoughtful review of our manuscript entitled “Use of Measles and Rubella Rapid Diagnostic Tests to Improve Measles and Rubella Detection and Targeting of Vaccination”, submitted to the Vaccines Special Issue “A World without Measles and Rubella: Meeting the Regional Elimination Targets on the Path to Global Eradication”. We appreciate the time and effort you have dedicated to providing this helpful feedback. Below, we address each of your comments and suggestions:

Comments and Suggestions for Authors

1. Measles and rubella are exanthematous virus infections that cause considerable morbidity and mortality worldwide, despite the availability of effective immunisation for close to 50 years. Contributory factors are the reduced uptake of immunisation due to individual choice or due to disruption of national immunisation programmes. Measles and rubella present with coryzal symptoms, fever, conjunctivitis and a maculopapular skin rash, that can be difficult to differentiate on clinical grounds alone from other viral diseases presenting with similar clinical features. Therefore, laboratory testing to confirm the diagnosis is important to monitor the effectiveness of routine immunisation programmes and also implement supplementary immunisation activities to deal with outbreaks.

Currently, laboratory-based enzyme immunoassay testing of sera for measles and rubella IgM antibodies is the mainstay of testing, but there are logistical problems in achieving these results in a timely manner. Genotyping of these two viruses are also available but have similar logistical disadvantages. Lateral  flow technology-base, rapid diagnostic tests have made a huge impact on the control of malaria, COVID-19 and HIV. Similar approaches would be highly valuable for the control of measles and rubella. There are commercial rapid diagnostic tests for detecting rubella IgM but subject to limited validation. 

This report highlights the development of rapid diagnostic tests for anti-measles IgM which appears to have promise. This communication cites some information about the currently available validation of rapid diagnostic tests to detect measles IgM. However, this manuscript does not provide a systematic review of the available rapid diagnostic tests for measles and rubella and their reliability and effectiveness under field conditions. 

Thank you for your comment. This manuscript was not intended to be a systematic review, but rather, was an invited manuscript meant to examine the current status of measles and rubella IgM RDT development, their potential impact on measles and rubella elimination efforts, and to reflect on current challenges related to their deployment and ways forward. To date, no commercially validated RDTs are available for measles or rubella rapid detection. The one promising measles IgM test in development has only been described in five published manuscripts, three of which are laboratory evaluations using stored sera, one recently published field trial from Malaysia, and one review from 2020. There are also a small number of unpublished trials such as those mentioned in the manuscript in Uganda and India. While we had provided some information around the performance characteristics described in each of these laboratory and field trials, we agree that this could be further expanded upon. We have now expanded this section to be more comprehensive and also include additional information on viral RNA extraction and detection from test devices from these published studies.

To lines 162-201 we have added “The test was evaluated using 170 serum specimens collected through measles surveillance or vaccination programs in Ethiopia, Malaysia, and the Russian Federation. An additional 282 oral fluid specimens from the measles, mumps, and rubella (MMR) surveillance program of the United Kingdom of Great Britain and Northern Ireland were also screened, with the Microimmune measles IgM capture EIA (Microimmune Ltd, Hounslow, England) as the gold standard for comparison [31]. With serum specimens, the RDT had a sensitivity of 91% (69/76) and a specificity of 94% (88/94); with oral fluid, the sensitivity was 90% (63/70) and the specificity was 96% (200/208)[31].

The study also demonstrated that the RDT could support measles virologic surveillance, as viral RNA can be extracted from test devices, amplified by standard RT-PCR and used for sequence analysis to determine genotype at national or regional laboratories. Here, a panel of 24 oral fluids was used to investigate whether measles virus haemagglutinin (H) and nucleocapsid (N) genes could be amplified by polymerase chain reaction directly from used RDTs. Both H and N genes were reliably detected and the N genes could be sequenced for genotyping. Measles virus genes could be recovered from RDT test strips after storage for 5 weeks at 20–25 °C [31]. A subsequent evaluation of 103 suspected measles cases during a 2010 nationwide measles outbreak in Zimbabwe confirmed previous findings that it was possible to amplify measles genes from oral fluid using RDT test strips and identify the infecting measles genotype and strain. During the evaluation, there was no loss in the sensitivity of detection of the measles H gene from RDT strips compared to direct detection from oral fluid samples [32].

With support from the Bill & Melinda Gates Foundation (BMGF), this prototype RDT was redesigned into a more robust, cassette-housed device, requiring fewer procedural steps to generate a result, with demonstrated high sensitivity (95%) and specificity (96%) when evaluated on stored sera against the Enzygnost® anti-measles virus IgM EIA reference assay (Siemens Healthcare GmbH, Marburg, Germany) [33, 34]. In unpublished field trials in India and Uganda, the sensitivity of measles IgM detection in capillary blood using the RDT was comparable to published performance with RDT and to current laboratory-based EIA testing; sensitivity was slightly lower using oral fluid [34]. 

In a subsequent 2019 field trial in Malaysia, the sensitivity of the measles IgM RDT was lower than reported in earlier evaluations and measles endemic settings. RDT sensitivity using capillary blood was 43% and specificity was 98%; using oral fluid, sensitivity was 26% and specificity was 97% [35]. However, parallel measles and rubella vaccination and mop-up campaigns that occurred within the 8–56 days preceding specimen collection may have led to potential vaccine responses in some cases, contributing to discordant results. Additionally, measles incidence in Malaysia decreased between the study’s design and implementation phases, ultimately resulting in decreased statistical power. These data highlight the need to further evaluate both the performance and potential role of IgM RDT in low-incidence elimination settings [35].

2. “Here, we review the status of RDTs for measles and rubella Immunoglobulin M (IgM) testing, as well as ongoing questions regarding operational use and deployment of RDTs as part of global measles and rubella surveillance. Researchers at UK Health Security Agency (UKHSA, formerly Public Health England [PHE]) developed an RDT for measles IgM antibodies in 2011 [29]. With serum specimens, the RDT had a sensitivity of 91% (69/76) and a specificity of 94% (88/94); with oral fluid, the sensitivity was 90% (63/70) and the specificity was 96% (200/208) [28]. With support from the Bill & Melinda Gates Foundation (BMGF), this prototype RDT was redesigned into a more robust, cassette-housed device, requiring fewer procedural steps to generate a result, with demonstrated high sensitivity (95%) and specificity (96%) when evaluated on stored sera against the Enzygnost® anti-measles virus IgM EIA reference assay (Siemens Healthcare GmbH, Marburg, Germany) [30]. Additional studies have demonstrated that RDTs can also support measles virologic surveillance, as viral RNA can be extracted from test devices, amplified by standard RT-PCR and used for sequence analysis to determine genotype at national or regional laboratories [31]. 

However, in a 2019 field trial in Malaysia, the sensitivity of the measles IgM RDT was lower than reported in earlier evaluations and measles endemic settings. RDT sensitivity using capillary blood was 43% and specificity was 98%; using oral fluid, sensitivity was 26% and specificity was 97%, highlighting the need to further establish the role of the device in elimination settings.”

From the information given, the reader cannot judge the quality of the evidence provided, without having to review the original manuscripts cited. For example, there is no critical analysis of why the 2019 field trial in Malaysia had poor sensitivity using the rapid diagnostic test for anti-Measles IgM and whether since that time efforts have been made to improve the performance of these assays under field conditions. 

Thank you for this comment. We agree that additional detail and analysis should be included for the poor sensitivity identified in the 2019 field trial in Malaysia. To lines 192-201 we have added “In a subsequent 2019 field trial in Malaysia, the sensitivity of the measles IgM RDT was lower than reported in earlier evaluations and measles endemic settings. RDT sensitivity using capillary blood was 43% and specificity was 98%; using oral fluid, sensitivity was 26% and specificity was 97% [32]. However, parallel measles and rubella vaccination and mop-up campaigns that occurred within the 8–56 days preceding specimen collection may have led to potential vaccine responses in some cases, contributing to discordant results. Additionally, measles incidence in Malaysia decreased between the study’s design and implementation phases, ultimately resulting in decreased statistical power. These data highlight the need to further evaluate both the performance and potential role of IgM RDT in low-incidence elimination settings [32].”

It is not clear what additional contribution to this topic is made in this manuscript compared to the cited paper from 2020: https://doi.org/10.1016/j.coviro.2020.05.007. This manuscript appears to have two objectives. First, is to critically review the current scientific validity of rapid diagnostic tests for measles and rubella. Secondly, it tries to make a health-economic case for employing these techniques for measles and rubella surveillance on a global scale. In my opinion, it does not achieve either of these objectives, sufficiently to justify publication in the current format. The options for improving this manuscript are to either carry out a systematic review of currently available conventional and rapid diagnostic tests for measles and rubella, possibly in comparison with other similar viral illness. An alternative is to carry out a review of the health-economic case for using rapid diagnostic tests which could be achieved in a much shorter manuscript length.

Thank you for your feedback regarding the manuscript objectives. We agree that this manuscript does have two primary objectives, which are: 1). To describe the current status of measles and rubella IgM RDT development and performance and their potential impact on measles and rubella elimination efforts, and 2). Describe the ongoing challenges related to RDT deployment in countries, their development and financing challenges, and to reflect on potential ways forward. Given the limited number of both published and unpublished information currently available on these topics, we feel we have achieved these objectives in a manner that is both clear and useful to our target audience.

This manuscript is not intended to be a systematic review, however, we do agree that both objectives could be further expanded upon and the manuscript better structured to more effectively communicate our two aims. In addition to restructuring and adding additional subheadings throughout the manuscript, we have also expanded upon multiple sections, including “Current Laboratory Testing Methods” (lines 103-118), “Measles IgM Rapid Diagnostic Tests” (lines 155-215), “Integration of RDTs into Measles and Rubella Surveillance Programs” (lines 230-267), “Financing Challenges for Measles and Rubella IgM RDT Deployment” (lines 271-309) and “Validation of Commercially Available Measles and Rubella IgM RDTs and Guidance on Deployment” (lines 311-373). With regards to validation, deployment and financing challenges, we have added a number of edits, including a reference to the recently published target product profiles (TPPs) for both measles and rubella RDTs and EIAs by Foundation for Innovative New Diagnostics (FIND) (lines 329-329) (https://www.finddx.org/wp-content/uploads/2024/06/20240613_tpp_measles_rubell_FV_EN.pdf), as well as some further commentary on how rapid diagnostics for other diseases have been shown to improve early case detection, time to response, and reduce the economic costs of treatment (lines 351-365).

Additionally, while we agree that both the cited paper noted above (https://doi.org/10.1016/j.coviro.2020.05.007) and this manuscript describe the performance and operational challenges related to measles IgM RDT integration into national measles and rubella surveillance programs, we feel it important to provide newly available updates since its publication four years ago. This includes updates on measles RDT development, such as current stakeholders and the recent publication of the RDT and EIA TPPs by FIND, country use cases (specifically, the recently published Malaysia field trial where the RDT sensitivity was significantly lower than noted in previous evaluations), and integration of both quality management systems and reporting results from RDT testing into routine health information management systems at local and national levels. We also discuss current updates regarding rubella IgM RDTs, and the potential implications of integrating both RDT tests into national measles and rubella surveillance programs. The second objective of this manuscript (to describe the ongoing challenges related to RDT deployment in countries, development and financing challenges, and to reflect on potential ways forward) is not discussed in any significant detail in the 2020 manuscript mentioned above, however, given the current barriers related to country deployment and financing, we felt this was important to discuss in parallel with these RDT development and performance updates, including providing some reflection on the potential ways forward.

Given a systematic reviews of commercially available serological tests for the detection of measles and rubella has been recently published (https://journals.asm.org/doi/pdf/10.1128/jcm.01339-23) and there are currently no studies evaluating the health-economic impact of using rapid diagnostic tests for measles and rubella detection and response for review and comparison, we do not feel these are the most appropriate topics to focus on for this manuscript at this time. We do agree that studies assessing the cost and effectiveness of using the measles RDT compared to the current status quo (EIA with serum), and evaluation of these programmatic cost implications are critical to undertake in the future as MR RDTs become available. 

Thank you once again for your valuable feedback. We hope the revisions address your concerns. We have also attached the revised tracked version of the manuscript for you to view.

Best regards,

Audrey Rachlin, PhD, MSc

Epidemiologist

Global Immunization Division

Centers for Disease Control and Prevention (CDC)

Reviewer 3 Report

Comments and Suggestions for Authors

Introduction

The introduction is clear and well arranged.

Methodology

The methodology must be improved. I suggest authors insert if it is possible a method of sources was used for the present review. Regardless of the type of narrative review, authors should clearly describe how analyses were conducted and provide justification for their approach and the aims of the review. Concentrate all laboratory methods in one table to be more visible to readers.

Discussion

 The main target for the authors is to improve how the screening with (RDTs) could help the vaccinations strategies. Both IgM and IgG antibodies are produced during the primary immune response and measles-specific IgM can be detected in the serum as early as the first day of rash onset. Following measles virus infection in an unvaccinated individual, measles IgM antibodies appear within the first few days (1–4 days) of rash onset, peak within the first week post rash onset and are rarely detected after 6–8 weeks. I think for areas such a Sub-Saharan Africa the main target it’s to increase the vaccination coverage and not the detection of pathogen. This approach of (RDTs) is more close to in early case identification, surveillance and case management and not for vaccination strategies. According to CDC (https://www.cdc.gov/measles/data-research/index.html ) with Updated on May 24, 2024, only 5% of the  142 measles cases were reported in USA was completed vaccination with 2 doses. However, to enhance the manuscript suggested to compare previous studies for other diseases and how (RDTs) help to improve the Detection and Targeting of Vaccination.

Author Response

Reviewer 3

Dear Reviewer,

Thank you for your thoughtful review of our manuscript entitled “Use of Measles and Rubella Rapid Diagnostic Tests to Improve Measles and Rubella Detection and Targeting of Vaccination”, submitted to the Vaccines Special Issue “A World without Measles and Rubella: Meeting the Regional Elimination Targets on the Path to Global Eradication”. We appreciate the time and effort you have dedicated to providing this helpful feedback. Below, we address each of your comments and suggestions:

Comments and Suggestions for Authors

Introduction

The introduction is clear and well arranged.

Thank you for this comment.

Methodology

The methodology must be improved. I suggest authors insert if it is possible a method of sources was used for the present review. Regardless of the type of narrative review, authors should clearly describe how analyses were conducted and provide justification for their approach and the aims of the review. Concentrate all laboratory methods in one table to be more visible to readers.

Thank you for this suggestion. We agree that is critical that methods for systematic reviews and meta-analyses, which utilize a prespecified method to synthesize findings from similar studies, should be clearly described within a publication. However, the objectives of this manuscript were not to provide a systematic review of measles and rubella RDTs, but rather to provide an overall framework on the current status of measles and rubella IgM RDTs by examining their development, impact on measles and rubella elimination efforts, and to reflect on current challenges related to their deployment and potential ways forward. There are currently only five publications that examine measles IgM RDTs in detail (none are currently available for rubella IgM RDTs), thus, no inclusion, exclusion, or prescribed search criteria/keywords were used when collating these findings. Additionally, MDPI Journal guidelines for Review submissions did not specify the requirement to include a methods section MDPI | Article Types.

However, we do agree that our current objectives could be better defined and the manuscript better structured to more effectively communicate our primary aims to the target audience. We have further added to the Introduction (lines 53-57) “Here, we review the status of RDTs for measles and rubella Immunoglobulin M (IgM) testing, as well as ongoing questions regarding the operational use and deployment of RDTs as part of global measles and rubella surveillance. We also discuss RDT financing challenges in low-and-middle-income countries and reflect on the way forward.” We have also added additional subheadings throughout the manuscript to better align with these objectives for additional clarity.

Discussion

The main target for the authors is to improve how the screening with (RDTs) could help the vaccinations strategies. Both IgM and IgG antibodies are produced during the primary immune response and measles-specific IgM can be detected in the serum as early as the first day of rash onset. Following measles virus infection in an unvaccinated individual, measles IgM antibodies appear within the first few days (1–4 days) of rash onset, peak within the first week post rash onset and are rarely detected after 6–8 weeks. I think for areas such a Sub-Saharan Africa the main target it’s to increase the vaccination coverage and not the detection of pathogen. This approach of (RDTs) is more close to in early case identification, surveillance and case management and not for vaccination strategies. According to CDC (https://www.cdc.gov/measles/data-research/index.html ) with Updated on May 24, 2024, only 5% of the  142 measles cases were reported in USA was completed vaccination with 2 doses. However, to enhance the manuscript suggested to compare previous studies for other diseases and how (RDTs) help to improve the Detection and Targeting of Vaccination.

Thank you for this suggestion. We agree that including additional studies on other diseases related to how RDTs can help to improve the detection timeliness and economic costs would be beneficial to our argument. To lines 349-363 we have added “Data on the cost-benefit and effectiveness of measles and rubella RDTs are also needed to make further investment decisions regarding diagnostics development to provide an economic case for future investment in RDTs. There have been estimates of the costs of RDTs for various diseases such as HIV, malaria, and tuberculosis [48-50] that have indicated the decentralization of testing using RDTs is likely to be cost-saving or cost-effective in resource-limited settings [51] and the fast turnaround time of RDTs resulted in a reduction in unnecessary clinical treatment costs, leading to saving for both patients and clinics [52]. While previous literature has also documented the costs of various aspects of measles and rubella surveillance [53], there is limited information available regarding the costs of implementing the measles RDT as part of a national measles surveillance program. Such data would help inform decision-makers and public health planners on the magnitude of resources required for introducing measles and rubella RDTs into the national surveillance program, how these costs may vary across geographies (e.g., rural vs. urban, endemic vs. elimination setting), and how this could impact the public health response for measles outbreaks.”

Thank you once again for your valuable feedback. We hope the revisions address your concerns. We have also attached the revised tracked version of the manuscript for you to view.

Best regards,

Audrey Rachlin, PhD, MSc

Epidemiologist

Global Immunization Division

Centers for Disease Control and Prevention (CDC)

Round 2

Reviewer 3 Report

Comments and Suggestions for Authors

Thank you for the response.